# In Vitro Study of Probiotic, Antioxidant and Anti-Inflammatory Activities among Indigenous *Saccharomyces cerevisiae* Strains

**DOI:** 10.3390/foods11091342

**Published:** 2022-05-05

**Authors:** Gabriella Siesto, Rocchina Pietrafesa, Vittoria Infantino, Channmuny Thanh, Ilaria Pappalardo, Patrizia Romano, Angela Capece

**Affiliations:** 1Scuola di Scienze Agrarie, Forestali, Alimentari ed Ambientali, Università degli Studi della Basilicata, Viale dell’Ateneo Lucano 10, 85100 Potenza, Italy; gasiesto1@virgilio.it (G.S.); angela.capece@unibas.it (A.C.); 2Dipartimento di Scienze, Università degli Studi della Basilicata, Viale dell’Ateneo Lucano 10, 85100 Potenza, Italy; vittoria.infantino@unibas.it (V.I.); ilaria.pappalardo@unibas.it (I.P.); 3Institute of Technology of Cambodia (ITC), Russian Federation Blvd, P.O. Box 86, Phnom Penh 12101, Cambodia; channmunythanh@gmail.com; 4Dipartimento di Economia, Universitas Mercatorum, 00186 Roma, Italy; patrizia.romano@unimercatorum.it

**Keywords:** probiotic yeasts, *Saccharomyces cerevisiae*, indigenous strains, health host, antioxidant activity, anti-inflammatory activity

## Abstract

Nowadays, the interest toward products containing probiotics is growing due to their potential health benefits to the host and the research is focusing on search of new probiotic microorganisms. The present work was focused on the characterization of indigenous *Saccharomyces cerevisiae* strains, isolated from different food matrixes, with the goal to select strains with probiotic or health-beneficial potential. A preliminary screening performed on fifty *S. cerevisiae* indigenous strains, in comparison to a commercial probiotic strain, allowed to individuate the most suitable ones for potential probiotic aptitude. Fourteen selected strains were tested for survival ability in the gastrointestinal tract and finally, the strains characterized for the most important probiotic features were analyzed for health-beneficial traits, such as the content of glucan, antioxidant and potential anti-inflammatory activities. Three strains, 4LBI-3, LL-1, TA4-10, showing better attributes compared to the commercial probiotic *S.*
*cerevisiae* var. *boulardii* strain, were characterized by interesting health-beneficial traits, such as high content of glucan, high antioxidant and potential anti-inflammatory activities. Our results suggest that some of the tested *S. cerevisiae* strains have potential as probiotics and candidate for different applications, such as dietary supplements, and starter for the production of functional foods or as probiotic to be used therapeutically.

## 1. Introduction

Probiotics are defined by the World Health Organization (WHO) and the Food and Agriculture Organization of the United Nations (FAO) as “live microorganisms which, when administered in adequate amounts, confer a health benefit on the host” [1]. The term has been also applied to the supplements that are added to animal feed in order to improve the health of the animal for the beneficial effects on their gut microbiota.

In the last decades, the use and the request of the commercial probiotics are growing in consequence of the worldwide interest to their positive impact on the human health. In fact, many clinical data confirm therapeutic effects of the probiotics in the treatment of gastrointestinal diseases (irritable bowel syndrome, gastrointestinal disorders, elimination of *Helicobacter pylori*, diarrhoeas) [2] and prophylaxis of respiratory and urogenital infections [3]. Probiotics play a significant role towards allergic reactions as atopic dermatitis, and in the regulation of common metabolic diseases (obesity, insulin resistance syndrome, type 2 diabetes) [4,5]. Scientific studies also report their anti-cancer activity and the enhance of the body’s immune response (immunomodulation) by the activation of specific genes of localized host cells [6]. Probiotics are able to modulate the composition of intestinal microbiota, by competition with the native microflora [7] and prevent the growth of pathogenic bacteria [8]. Nowadays, due to these health benefits, probiotics are more demanded each day and usually, they are taken orally, as drugs or through food [9].

To be included into the probiotic category, microorganisms must be selected on the basis of certain criteria, such as growth at the body temperature (37 °C), antibiotic resistance [10,11], survival at the unfavourable human intestinal environment (e.g., digestive enzymes, gastric and pancreatic juices, low pH), adhesion to gut epithelial cells and high surface hydrophobicity [12].

It is well known that the most probiotics currently commercialised are of bacterial origin belonging mainly to the genera *Lactobacillus* and *Bifidobacterium* [1,13], while yeasts are poorly studied for their probiotic aptitude [14,15]. *Saccharomyces cerevisiae* var. *boulardii* (Sb) is the only yeast recognized and actually marketed as probiotic [16]. Yeasts are widely distributed in food industry where are used as starters for the production of fermented food and beverages as bread, table olives, wine and beer [17,18].

Among others, *S. cerevisiae* is the most common yeast for food fermentation possessing the Qualified Presumption of Safety (QPS) status [19] assigned by European Food Safety Authority. Many researchers focused on both *Saccharomyces* and non-*Saccharomyces* yeast strains [20,21,22,23,24] as a good alternative to probiotic bacteria since they could exert a novel prophylactic or therapeutic effects towards the host by being part of the gut microbiota [25,26,27]. Immune system stimulation, degradation and elimination of bacterial toxins due to yeast’s protease activity or inhibition of bacteria adherence to gastrointestinal epithelial cells are some of the potential mechanisms used by yeasts to protect the host against pathogen microorganisms [20,28].

Furthermore, yeasts could be considered probiotic organisms because they contain various immune-stimulant compounds (e.g., β-glucans, proteases and mannanoligosaccharides) and they are not affected by anti-bacterial compounds [29,30]. Yeast β-glucan can play an important role as marker of intestinal inflammation and oxidative stress. Studies carried out in vivo by animal models confirmed the health benefits associated with β-glucan [31] and it is demonstrated the effect of *S. cerevisiae*-derived β-glucan on mouse macrophage polarization. Macrophage are leukocytes involved in innate immunity and inflammation and can be activated by different signals. Yeast probiotics exert an immunomodulatory activity including effects on M1 macrophage function [32]. Martins et al. [33] demonstrated that *S. cerevisiae* var. *boulardii* reduces levels of pro-inflammatory cytokines and NF-κB signaling upon *Salmonella enterica* serotype Typhimurium infection, thus showing anti-inflammatory properties.

Activated macrophages (M1) triggered by pathogen-associated molecular patterns (PAMPs) like lipopolysaccharide (LPS) acquire an inflammatory phenotype and are characterized, among others, by generation of reactive oxygen species (ROS) and nitric oxide (NO). It is well known the critical role of LPS in host-pathogen interaction with the innate immune cells. By binding to the toll-like receptor 4, LPS triggers a strong inflammatory response. This phenotypic change of activated macrophages is mediated by a significant metabolic and gene expression reprogramming which lead to the production of a wide range of inflammatory mediators [34,35].

During this shift from resting to activated status, a metabolic reprogramming occurs to cope with the energetic needs of cells and in turn to achieve the specific function. About that, the Warburg effect—a well-known hallmark of cancer metabolic changes—has been described in M1 macrophages too. In both activated macrophages and cancer cells, this metabolic rewiring supplies building blocks for the increased biosynthetic demand thus driving toward the specific phenotype [36,37]. In this context, rate of pentose phosphate pathway together with mitochondrial citrate export are increased in order to produce NADPH needed for the synthesis of NO and ROS inflammatory mediators [38,39,40].

Considering the lack of knowledge regarding the probiotic properties of indigenous yeasts, the aim of this study was the characterization of indigenous *S. cerevisiae* strains, isolated from different food matrixes, with the goal to select strains with probiotic or health-beneficial potential. At this purpose, a preliminary screening was performed on fifty *S. cerevisiae* indigenous strains, isolated from different food matrixes, in order to evaluate their probiotic aptitude as tolerance to gastrointestinal conditions (pH, temperature), antibiotic resistance and hydrophobicity. Fourteen selected strains were tested for their behaviour after simulated transit into gut and only three strains (4LBI-3, TA4-10, LL-1) in comparison to the probiotic commercial strain Sb, were analyzed to evaluate functional activities, such as the content of glucans, antioxidant and anti-inflammatory activities.

## 2. Materials and Methods

### 2.1. Origin of Yeast Strains

In this work, fifty indigenous *S. cerevisiae* strains (non-commercial strains), belonging to UNIBAS Yeast Collection (UBYC), University of Basilicata (Potenza, Italy) and previously isolated from food environments, were studied (Table 1). Furthermore, two commercial strains, EC1118 (Lallemand Inc., Toulose, France), a winemaking starter, and a probiotic *S. cerevisiae* var. *boulardii* strain (coded as Sb) and isolated from the commercial product Codex (Zambon, Italy), were used. The strains were maintained at −80 °C in a glycerol solution (Sigma, Milan, Italy); before each experiment, yeast strains were grown in YPD (yeast extract 1%, bacteriological peptone 2%, glucose 2%, agar 2%, Oxoid, Hampshire, UK) at 26 ± 2 °C for 24 h.

### 2.2. Preliminary Characterization

#### 2.2.1. Growth at Human Body Temperature

This test was performed in YPD-agar medium; each strain was inoculated as spot with 5 µL of cell suspension with a concentration of around 10^6^ cells/mL, measured at 600 nm by spectrophotometer (SPECTROstar Nano, BMG Labtech, Ortenberg, Germany). Plates were aerobically incubated at 37 ± 2 °C and 26 ± 2 °C (as positive control) for 48 h. Cell growth level was evaluated by comparing the size and thickness of each strain spot. All the experiments were performed in duplicate.

#### 2.2.2. Strain Tolerance to Different pH

This experiment was performed by inoculating sterile tubes containing 10 mL of YPD broth, adjusted to different pH (2.5, 3.0, 3.5, 4.0, and 7.2) with approximately 4.8 × 10^6^ cells/mL of each strain and incubated at 37 ± 2 °C. Positive controls were obtained by inoculating the different yeasts in non-modified pH YPD-broth (pH = 6.55). Yeast growth was evaluated after 24 of incubation as absorbance at 600 nm. The analyses were performed in duplicate and data were elaborated as Growth Index (GI) following Bevilacqua et al. [51]
GI = (Abss/Absc) × 100%
where Abss is the absorbance of the samples at different pH, whereas Absc is the absorbance of the positive control. GI values were classified as follows:GI < 50%: Yeast Inhibition50% < GI < 75%: Partial InhibitionGI > 75%: Growth Similar to the Control

#### 2.2.3. Hydrophobicity of the Cells Surface

Hydrophobicity was determined as the ability of the yeast strains to adhere to hydrocarbons and it was expressed as Microbial Adhesion to Solvents (MATS), following the protocol reported by Chelliah et al. [52], with some modifications. Yeast cultures in stationary phase (inoculated in YPD broth for 48 h) were concentrated by centrifugation at 5000 rpm for 10 min, washed twice with PBS (Potassium phosphate buffer 0.1 M, pH 7.4, Merck, Darmstadt, Germany) and suspended again in 3 mL of the same buffered solution. The OD at 600 nm of the suspension was adjusted with PBS (pH 7.4) to a value of 1.0 (A0). 0.2 mL of toluene (Merck, Darmstadt, Germany) was added to 1 mL of this suspension (1 × 10^7^ cell/mL) and mixed for 2 min. After 1 h of incubation at 30 ± 2 °C, the aqueous phase was removed from the upper solvent layer containing the cells and the OD was determined again at 600 nm (A1). The MATS was calculated as:MATS = [(A0−A1)/A0] × 100%

#### 2.2.4. Antibiotic Resistance

Antibiotic-resistance was carried out with the Kirbye Bauer method according to the NCCLS protocol [53]. Five different antibiotics were used: streptomycin (ST), chloramphenicol (CH), gentamicin (GE), erythromycin (ER), and oxytetracycline (OX) (Sigma, Milan, Italy). Tests were performed on YPD agar added with five different concentrations of each antibiotic: 40, 80, 100, 130 and 150 µg/mL, in comparison to the control without antibiotic addition. Cell suspensions were adjusted to 0.2 of Optical Density (OD) (10^6^ cells/mL), and 8 µL of each strain were dropped on the antibiotic-YPD plates, which were incubated at 37 ± 2 °C for 48 h. Cell survival and growth level on the different concentrations of the five antibiotics were qualitatively evaluated, by comparing the size and thickness of each strain spot with the corresponding control. All the experiments were performed in duplicate.

### 2.3. Screening for Potentially Probiotic Traits: Simulation of the Transit into Gut

The experiment was performed according the method of Picot and Lacroix [54], with some modifications. Yeasts were grown in YPD broth for 48 h, then cells were washed twice and suspended into sterile distilled water. For this analysis, in order to simulate the sequential digestion, two different solutions were prepared:▪ Gastric Juice: 0.26 g/L of pepsin (porcine gastric mucosal, Sigma, Milan, Italy) were dissolved in HCl 0.1 N and the final pH of the solution was adjusted to 1.9 with 1 N NaOH;▪ Intestinal fluid: 1.95 g/L of pancreatin (porcine pancreas, Sigma, Milan, Italy) were dissolved in a phosphate buffer solution (NaH_2_PO_4_/Na_2_HPO_4_ 0.02 M, pH 7.5), and added with 3 g/L of bile extract (bile extract porcine, Sigma, Milan, Italy), dissolved in sterile distilled water; and the final pH was adjusted to 7.5 with 1N NaOH.

Each strain, at a concentration of about 2.0 × 10^7^ cells/mL, was inoculated in 50 mL of sterile distilled water for 15 min at 4 °C. Then, the pH was adjusted to 1.9 with HCl 1 N and the volume was diluted to 60 mL with distilled water; subsequently, 20 mL of the pepsin solution were added and the samples were incubated at 37 ± 2 °C for 3 h (simulation of the stomach transit). Every 30 min, an aliquot of sample (2 mL) was picked up and kept in ice before the inoculation on the plates. After 3 h, the reaction was stopped by increasing the pH to 7.5 with 1 N NaOH, and added with 5 mL of a concentrated sodium phosphate buffer and 2 mL of the bile salt. The pH was adjusted to 7.5 with 1 N NaOH and distilled water was added to obtain a final volume of 90 mL; after that, 10 mL of the pancreatic solution were added and incubated at 37 ± 2 °C for 24 h to simulate the intestinal transit. Every 30 min, a sample of 2 mL was picked up and kept in ice before the spreading on the plates. All the experiments were performed in duplicate. Data were elaborated as Viability Index (VI):VI = (logNt/logN0) × 100%,
where Nt is yeast cell concentration for each time and N0 the initial inoculum.

### 2.4. Evaluation of Functional Activities on Four Selected Strains

Three indigenous strains, selected on the basis of previous screening (TA4-10, LL-1, 4LBI-3), in comparison to the probiotic strain Sb, were tested for evaluation of further biological activities, such as the content of glucans, antioxidant activity and cytotoxicity assay.

#### 2.4.1. Total Antioxidant Activity

The total antioxidant activity, evaluated on the basis of the scavenging activity of the 1,1-diphenyl-2-picrylhydrazyl (DPPH, Sigma, Milan, Italy) free radical, was determined by the protocol described by Chen et al. [14] and Capece et al. [49], with some variations. Briefly, the cells pellets, recovered by centrifugation (1200 rpm, 5 min) of 1 mL of yeast culture, grown on YPD broth for 24 h, was added with 1 mL of methanolic DPPH radical solution (0.2 mM in methanol, Sigma, Milan, Italy). The mix was vortexes and then incubated for 30 min at room temperature in darkness. The reaction tubes were centrifuged (1200 rpm, 5 min) and the absorbance at 517 nm was measured (in triplicate) on 300 μL of the supernatant. The blank used was pure methanol. The scavenging ability was defined as follows:Scavenging rate (%) = [1 − (A sample/A blank)] × 100%

#### 2.4.2. Determination of Glucans

Total glucans and β-glucans contents were analyzed by using a commercially available kit (Megazyme, Wicklow, Ireland), an enzymatic method based on acidic hydrolysis of the β-glucans, followed by enzymatic digestion to glucose [55]. For determination of total glucans, the yeast cells were aerobically grown in YPD broth at 30 °C for 24 h in order to reach the exponential phase of cultures. The yeast cultures were stopped at 2 × 10^7^ cells/mL and harvested by centrifugation at 4700 rpm for 10 min. Yeast pellets (about 100 mg) were suspended in 2.0 mL of 12 M HCl, stirred at 30 °C for 1 h to hydrolyse total glucans, following the manufacture’s instruction. In details, each tube was added with 10 mL of water and placed in a boiling water bath for 2 h, after that the tubes were cooled at room temperature and added with 200 mM sodium acetate buffer (pH = 5). An aliquot of the sample was centrifuged at 1500 rpm for 10 min and 0.1 mL of extracts was added with 0.1 mL of a mixture of exo-1,3-β-glucanase (20 U/mL) and β-glucosidase (4 U/mL) in 200 mM sodium acetate buffer. The mixture was incubated at 40 °C for 60 min, after that each tube was added with 3.0 mL of glucose oxidase/peroxidase (GOPOD) reagent and incubated at 40 °C for 20 min. The absorbance of all samples was measured at 510 nm against the blank.

The second step of enzymatic kits analysis was addressed to the determination of Alfa-Glucans (α-Glucans). In this step, 100 mg of yeast pellet were mixed with 2 mL of 2 M KOH; after stirring in ice-water, the samples were added with 8 mL of 1.2 M sodium acetate buffer (pH 3.8), 0.2 mL of amyloglucosidase (1630 U/mL) plus invertase (500 U/mL) and the mixture was incubated in a water bath at 40 °C for 30 min. After this step, the samples were centrifuged at 1500 rpm for 10 min and 0.1 mL of supernatants was added with 0.1 mL of sodium acetate buffer (200 mM, pH 5.0) plus 3.0 mL of GOPOD reagent and incubated at 40 °C for 20 min. In the last step, the absorbance of all samples was measured at 510 nm against the reagent blank. The β-Glucan content was calculated as difference between Total Glucans and α-Glucans by using the Megazyme Mega-CalcTM software.2.4.3. (Megazyme, Wicklow, Ireland) Cytotoxicity Assay and Anti-inflammatory Activity.

##### Cell Culture and Treatments

Human monoblasticleukemia U937 cell line (Interlab Cell Line Collection (ICLC) HTL94002) was grown in suspension in Roswell Park Memorial Institute (RPMI) 1640 medium (Sigma, Milan, Italy) supplemented with 10% (*v/v*) fetal bovine serum, 2 mM L-glutamine, 100 U/mL penicillin, and 100 μg/mL streptomycin at 37 °C in 5% CO_2_ as previously reported [56]. U937 cell were seeded in RPMI (CTRL) or in RPMI where the four strains were grown for 24 h. Pro-monocytic U937 cells were differentiated to macrophages by 10 ng/mL phorbol 12-myristate 13-acetate (PMA). U937/PMA cells, where indicated, were stimulated with 200 ng/mL of lipopolysaccharide from *Salmonella enterica* serotype Typhimurium (LPS) [57].

##### Cytotoxicity Assay

The effects of TA4-10, LL-1, Sb, 4LBI-3 on U937/PMA cell proliferation were determined using a Millipore Scepter™ handheld automated cell counter (Merck Millipore, Darmstadt, Germany) 72 h after incubation, according to the manufacturer’s instructions [58]. Briefly, cells were cultured into 96-well plates (5 × 10^4^ cells/well) in RPMI (CTRL) or in RPMI where the four strains were grown for 24 h. After 72 h, U937/PMA cells were collected in 1.5 mL microfuge tubes and counted.

##### ROS and NO Detection

To evaluate reactive oxygen species (ROS) and nitric oxide (NO) levels, U937/PMA cells were triggered by 200 ng/mL of LPS in the presence or absence of RPMI where the four strains were grown for 24 h. After 24 h, ROS and NO concentrations were measured by using 6-carboxy-2′,7′-dichlorodihydrofluorescein diacetate (DCF-DA, Thermo Fisher Scientific, San Jose, CA, USA) and -amino-5-methylamino-2′,7′-difluorofluorescein diacetate (DAF-FM diacetate, Thermo Fisher Scientific, San Jose, CA, USA), respectively, as previously described [59]. Briefly, cells were collected in tubes and counted. After centrifugation, the cell pellet was resuspended in PBS to have 10^5^ cells/100 μL. Then, 10 µM DCF-DA or 2.24 µM DAF-FM diacetate were added. After 30 min of incubation in the dark at 37 °C, a 100 μL portion of sample was transferred to triplicate groups of wells on a black microtiter plate and the fluorescence was revealed using a GloMax plate reader (Promega, Madison, WI, USA).

### 2.5. Data Analysis

Data regarding the strain tolerance to different pH and the hydrophobicity of the cell surface were analyzed by stacked chart. Statistical significance of differences between the strains for the simulation of the transit into gut, was determined using one-way analysis of variance (ANOVA) and the homogeneity of variance was verified applying the Levene’s test. Differences were considered as significant with a *p* value ≤ 0.05. Results regarding cytotoxicity assay and ROS and NO detection anti-inflammatory activity were shown as means ± SD of, at least, three independent experiments and were analyzed by one-way analysis of variance (ANOVA). Differences were considered as significant (*p* < 0.05), very significant (*p* < 0.01), and highly significant (*p* < 0.001).

Paleontological Statistics Software (PAST) ver. 3.26 (Natural History Museum-University of Oslo, Oslo, Norway) [60] was used for statistical analyses.

## 3. Results and Discussion

### 3.1. Preliminary Characterization

In the preliminary phase, fifty indigenous *S. cerevisiae* strains, and two commercial strains, EC1118 and Sb (as positive control strain) were characterized by different tests, such as growth at human body temperature, tolerance to different pH levels, hydrophobicity of the cell surface and antibiotic resistance.
-*Growth at human body temperature:* The temperature is an important physical parameter to evaluate the probiotic aptitude of the microorganism. In this study, the experiments were carried out at 37 °C, the typical temperature of the human body, and all the strains were able to grow at 37 °C (data not shown). These results agree with previous findings, reporting the ability of yeast strains to grow at human body temperature [22], although yeasts usually show a trend toward surviving to lower temperatures than lactic acid bacteria [61], and confirm that indigenous yeast strains might be an interesting source for the selection of new potential probiotic strains.-*Strain tolerance to different pH:* As regards the tolerance to pH, different pH values (2.5, 3.0, 3.5, 4.0, and 7.2) were investigated in order to evaluate the ability of tested yeasts to survive at the pH of the gastro-intestinal environment. In fact, the probiotics must survive harsh conditions during their passage through the intestinal tract [11]. All the strains exhibited a growth similar to the control (GI > 75%) to pH 3.5, 4.2, and 7.2, whereas the lowest pH values (2.5 and 3.0) influenced the yeast growth. In particular, at pH 2.5, 15 strains were inhibited (GI < 50%), 31 strains (59.61%) showed a partial inhibition (50% < GI < 75%), whereas only 6 strains exhibited a good tolerance to low pH (GI > 75%). At pH 3.0, 13 strains exhibited a partial inhibition (50% < GI < 75%) and 39 strains (75%) grew similar to the control (GI > 75%) (Figure 1). The highest tolerance to low pH, with a GI > 90%, were found in five strains, namely FD-3, M5-15 (both isolated from sourdough), TA4-10, CA10-4sc2 (both isolated from grape must) and 9-15G, isolated from honey. Furthermore, the majority of indigenous strains analyzed in this step was more tolerant to low pH level than Sb, the commercial probiotic strain. Resistance to acidic conditions was not surprising since the strains studied in this work were isolated from low pH environments, such as grape must, or substrate in which they coexist with lactic acid bacteria, such as sourdough [62].-*Hydrophobicity of the Cell Surface*: Hydrophobicity was defined as an interaction between microbial and host cells, mediated by cell-surface proteins and lipoteichoic acids [63]. As a consequence, this is one of the main criteria for the selection of strains with potential probiotic activity, because the microbial strain included in the probiotic category must have the ability to adhere to the intestinal mucosa for colonization and modulation of the immune system against pathogens [64]. The hydrophobic ability of the yeasts was reported in Figure 1. The results highlighted that only two strains (BP2-33, and SA10-19) showed 0% MATS (no hydrophobic activity), while the majority of the strains (35) exhibited a MATS in a range between 50% to 75%, which were much higher than the level found in *S. cerevisiae* var. *boulardii* strain used as control, confirming other results reporting better hydrophobicity level in indigenous strains than in control strains [65,66].

Four strains (4LBI-3, LL-1, P4 and TA4-10) showed the highest percentage of MATS (>75%).
-*Antibiotic Resistance*: The fifty-two strains were tested for the resistance to five antibiotics, at five different doses (40, 80, 100, 130, and 150 µg/mL) and all the strains survived in the presence of the highest dose of the tested antibiotics (150 µg/mL); in fact, no inhibition zone was observed around the spotted strains (data not shown). The high resistance of the *S. cerevisiae* strains to these antibiotics provide them advantage over bacteria in patients undergoing antibiotic treatment for therapeutic use [67]. Furthermore, this characteristic is of great interest also for potential application of these strains as probiotic sources in animal feed in consequence of potential presence of antibiotics residues in animal feed.

On the basis of the results obtained by preliminary characterization, 14 strains, exhibiting the best combination of the tested parameters, were chosen for further analysis; the characteristics of the selected strains are summarized in Table 2.

### 3.2. Screening for Potentially Probiotic Traits: Simulation of the Transit into Gut

In the second phase of the study, the 14 selected strains were tested for their potential probiotic activity. In this step, the 14 selected strains were tested for their survival level to the simulated transit into gut, by evaluating the tolerance level to gastric and pancreatic juices, which are important criteria in the selection of probiotic microorganisms, allowing them to survive during the transit across the human digestive tract.

The resistance to gastric juice was evaluated by calculating the Viability Index (VI, %) in Gastric Juice (GJ), simulating of the stomach conditions, after 0.5 h and 2 h of incubation (Figure 2). After 30 min, the SC5-25 strain exhibited the lowest VI (87.35% ± 0.92), whereas the highest VI (100%) was found in four strains (CA10-4sc2, EC1118, M4 and TA4-10). After 2 h, the exposure time to stomach acidic conditions recommended by different authors [22,68], the surviving level of the strains remained almost unchanged for the majority of the strains, whereas a slight increase of VI was observed for some of them, such as M5-15, Sb, SC5-25.

As regards the simulation of the intestinal transit, high variability among the analyzed yeasts was found for strain survival level to the simulated intestinal fluid (Figure 3). After 3 h, all the strains showed a VI > 50%, except the control strain Sb (45.36% ± 3.25) and CD2-6sc2 (47.89% ± 3.25). In particular, five strains exhibited a VI > 80% [EC1118 (84.80% ± 7.15), M5-15 (86.56% ± 7.55), P4 (89.05% ± 7.36) and TA4-10 (84.90% ± 7.63)] and LL-1 strain showed the highest VI (93.77% ± 8.06). After 24 h, a certain viability loss was observed for all the strains, except for EC1118 (84.80 to 98.30% ± 8.88), LL-1 (93.77 to 99.98% ± 8.95), M3-59 (from 77.61% ± 6.54 to 82.35% ± 7.14) and M4 (from 75.84% ± 6.39 to 85.76% ± 7.39). Two strains exhibited a viability loss of about half between 3 h and 24 h incubation, which were BA-215 (69.59% ± 5.43 to 38.84% ± 2.54) and P4 (89.05% ± 7.36 to 30.00% ± 2.94). These results showed the existence of high differences among the analyzed strains, confirming results previously reported by other authors that probiotic capacities are strain dependent [65,69,70]. The indigenous strains, characterized by higher resistance to digestive enzymes, such as pepsin, pancreatin as well as bile salts, than the commercial probiotic strain, demonstrating to be able to survive GI transit, can represent new microorganisms possibly affecting the microbial balance in the human body.

All the data regarding the screening of the 14 *S. cerevisiae* strains for potentially probiotic traits were statistically analyzed and used for the matrix plot reported in Figure 4. Three indigenous strains (4LBI-3, LL-1, TA4-10) were chosen on the basis of high tolerance to bile salts (VI > 75% after 3 h) and high survival level (VI > 95%) after the simulation of the transit into gut, as already reported (Figure 2 and Table 2). Commercial probiotic Sb was also used in the further characterization as reference strain, although this strain showed the lowest tolerance to bile salts, mainly after 24 h of incubation.

### 3.3. Evaluation of Functional Characteristics on Four Selected Strains

#### 3.3.1. Antioxidant Activity and Glucans Content

In the final step of this study, four strains were screened for functional characteristics other than probiotic traits, such as the antioxidant capacity. Several studies have reported the natural antioxidant capacity of yeast cells [14,62,71]. In our study, the in vitro antioxidant activities of the four chosen strain cells, measured as the scavenging effects on DPPH radicals, is reported in Figure 5. All of them showed antioxidant capacity, but at different level; the highest scavenging activity was exhibited by 4LBI-3 cells (49.08% ± 3.05), followed by TA4-10 cells; both the strains showed significantly higher antioxidant activity than the control strain. The strain LL-1 showed the lowest level (36.12% ± 2.47). However, all the four yeast strains showed antioxidant activity, at level comparable with the results obtained in other studies addressed to evaluate the health-beneficial potential of yeast strains isolated from food [22,70]. Although different mechanisms may be involved in antioxidant activity of yeast cells, it is believed that this functional activity is correlated to the content of (1/3)-β-D-glucan and other β-glucans found in the cell wall [72,73], other than cellular compounds, such as some antioxidant enzymes like superoxide dismutase, glutathione peroxidase and catalase [14]. In fact, the β-glucans were described as molecules with different biological activities that can include anti-inflammatory, antioxidative (in terms of free radical scavenging) and immunomodulation. Until now, few studies reporting the effect of source (or strain) of yeast as important factors affecting the β-glucan content are available [74,75]. The amounts of β-glucans extracted from the four selected strains is highly variable among them (Figure 5), demonstrating that the β-glucan content of a yeast is strain/species dependent [76,77]. Two indigenous strains showed a content significantly higher or similar (TA4-10 and 4LBI-3, respectively) than the content found in the commercial probiotic strain Sb, while LL-1 strain exhibited the lowest amounts of β-glucans (7.83 ± 0.68% *w/w*). Our results confirm the existence of a certain correlation between β-glucans content and antioxidant activity, although other component might influence the antioxidant activity of yeast cells as the strain exhibiting the highest scavenging ability on DPHH (4LBI-3) was not the same characterized by the highest content of β-glucans (TA4-10), as previously reported by other authors [14].

#### 3.3.2. Cytotoxicity Assay and Anti-Inflammatory Activity

First, we evaluated the effect of TA4-10, LL1, 4LBI-3 selected yeast strains and Sb control strain on U937/PMA cell proliferation by cell counting. Therefore, U937/PMA macrophage cells were cultured into 96-well plates in RPMI alone (CTRL) or in RPMI where the four selected strains were previously grown for 24 h. The determination of U937/PMA macrophage cell line after 72 h revealed that no strain inhibited U937/PMA cell proliferation (Figure 6). These results suggest that the three indigenous yeasts, similarly to the commercial probiotic strain, could be considered as non-toxic strains in consequence of their very low cytotoxicity.

#### 3.3.3. ROS and NO Detection

The next step was addressed to evaluate the potential anti-inflammatory activity of TA4-10, LL-1, 4LBI-3 and Sb strains. To this end, a gram-negative bacterial lipopolysaccharide (LPS), a major component of the outer membrane was used as inducer in U937/PMA macrophage cell line. The levels of two mediators closely linked to the inflammatory response, i.e., reactive oxygen species (ROS), specifically superoxide anion radical, and nitric oxide (NO), were assessed after LPS treatment in the presence or absence of RPMI, where the four strains were previously grown for 24 h. As shown in Figure 7A, the strains that significantly reduced ROS levels are, in order of intensity, LL-1, 4-LBI-3 and TA4-10. Conversely, the reference strain Sb does not seem able to counteract superoxide anion production in LPS-induced U937/PMA macrophage cell line (Figure 7A). Subsequently, the effect of yeast strains on NO production was tested in U937/PMA macrophage cell line triggered by LPS. Nitric oxide, a key biosignaling molecule produced in both peripheral tissues and the central nervous system, is synthesized from L-arginine to stoichiometric quantities of NO and L-citrulline using molecular oxygen and NADPH as cofactors by a family of enzymes known as nitric oxide synthases (NOSs). In LPS-induced U937/PMA macrophage cell line, TA4-10 and 4LBI-3 significantly reduced NO levels compared to LPS-activated macrophages by 25 and 14%, respectively (Figure 7B). Also in this trial, the strain less active in inhibiting NO production was the control strain Sb.

These results confirm the higher potentiality of indigenous strains to be used as potential probiotic than the commercial strain Sb. In fact, intact cells of indigenous strains were more able against LPS-induced inflammatory systems than intact cell of Sb. It’s widely reported the activation of the immune systems by β-1,3/1,6 glucans [78,79]. In our case, we can speculate a correlation between glucan content and anti-inflammatory activity only for the activity toward NO production as the strains possessing the highest ability to reduce the NO levels compared to LPS-activated macrophages were TA4-10 and 4LBI-3, which were the two strains characterized by the highest β-glucans content (Figure 5).

As regards the effect towards ROS, the highest efficacy was obtained for cells of LL-1 strain, characterized by the lowest β-glucans content, although the antioxidant activity of LL-1 cells was expressed at level comparable with other strains. These data confirm that, in addition to β-glucans, yeast cells contain a complex array of compounds, some of them also derived from cell walls, such as mannanoligosaccharides and glyco-proteins, other than peptides, amino acids, nucleotides, lipids and organic acids, with various capacities for driving pro-inflammatory and anti-inflammatory responses [80]. Some authors [81] found strong anti-inflammatory activity in stimulated human blood cells in β-glucan-depleted products produced by enzyme hydrolysis from brewer’s and baker’s yeasts.

## 4. Conclusions

The indigenous *Saccharomyces cerevisiae* strains tested in this study, mainly isolated from spontaneous fermented foods, exhibited a high biodiversity level for probiotic traits, confirming that indigenous yeasts represent an interesting source for the selection of new probiotic strains. Three strains, 4LBI-3, LL-1, TA4-10, showed promising probiotic activity; in particular, 4LBI-3 strain, isolated from spontaneous fermentation of Aglianico del Vulture grape must, the typical cultivar of Basilicata region (Southern Italy), might be considered the strain showing the best combination of probiotic and heathy traits. However, all the three *S. cerevisiae* strains, previously tested for parameters of technological interest, have the potential to be used as starter cultures for the production of functional fermented foods. As regards the therapeutic application, further research should be done to ensure safety and efficiency of these new potential probiotic yeasts.

## Figures and Tables

**Figure 1 foods-11-01342-f001:**
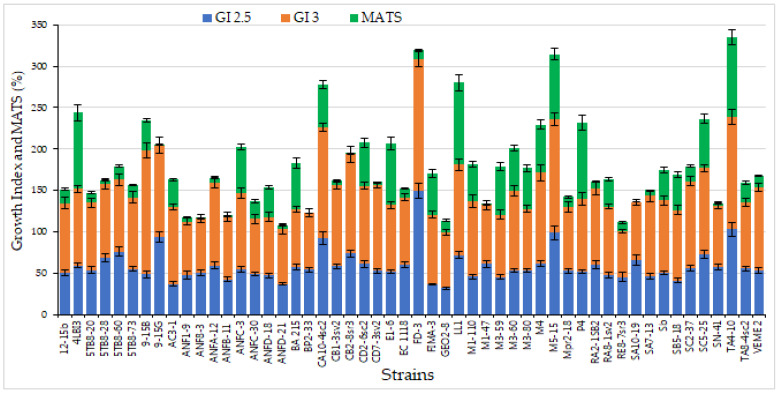
Stacked chart obtained by preliminary screening of 52 *Saccharomyces cerevisiae* strains based on the evaluation of growth index (GI) at low pH (2.5 and 3.0) after 24 h of incubation and hydrophobic properties GI was calculated as ratio between strain and positive control growth level, whereas hydrophobic ability was expressed as Microbial Adhesion to Solvents (MATS%), by using toluene as solvent.

**Figure 2 foods-11-01342-f002:**
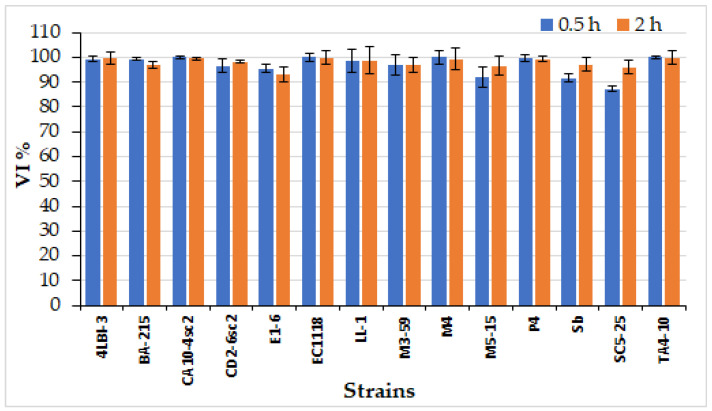
Viability index (VI) of the 14 chosen strains after exposure to simulated gastric juice after 0.5 and 2 h of incubation. The data are expressed in percentage (%) of the ratio between yeast cell concentration for each time and the initial inoculum and are reported as mean value ± SD of two independent experiments.

**Figure 3 foods-11-01342-f003:**
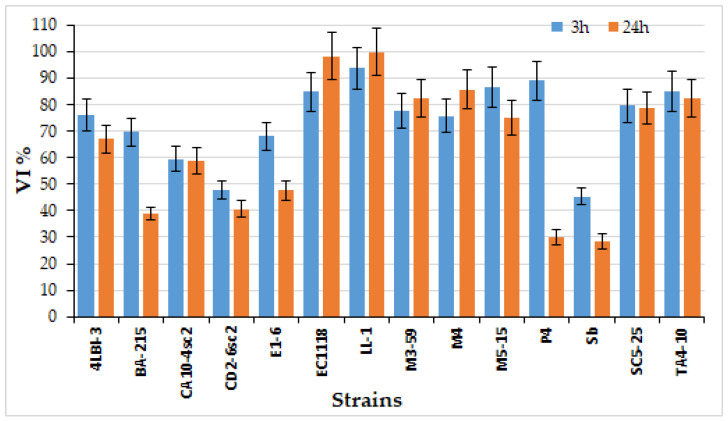
Strain survival after the exposition to simulated intestinal fluid after 3 h and 24 h of incubation. Data are expressed as VI (%) and reported as mean value ± SD of two independent experiments.

**Figure 4 foods-11-01342-f004:**
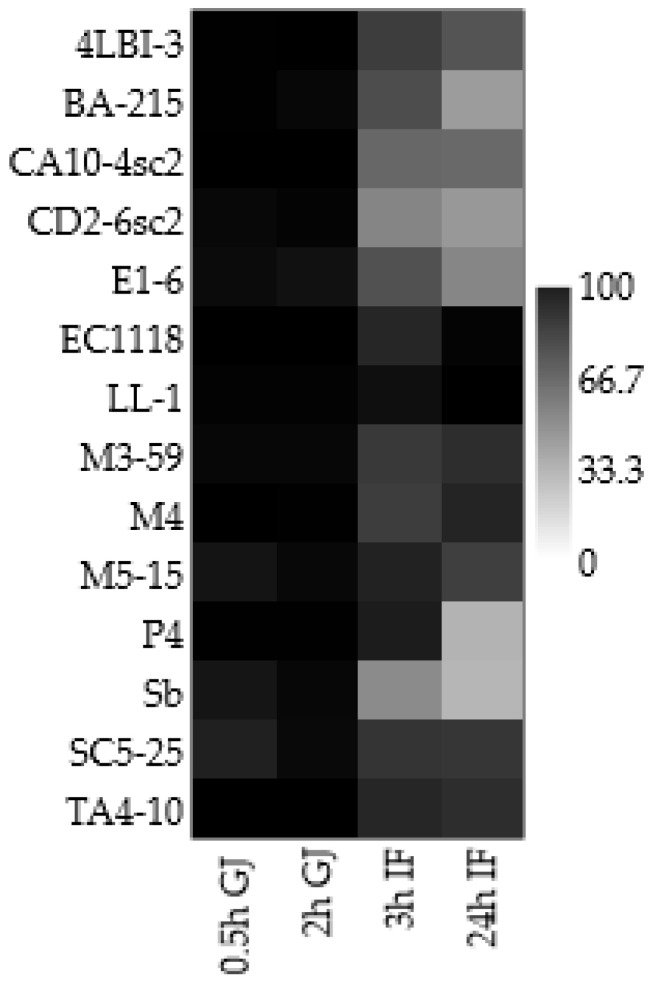
Matrix plot based on the evaluation of simulation of the transit into gut of the 14 *Saccharomyces cerevisiae* strains. The results are reported as Viability Index and expressed with a colour scale, where black indicate the highest Viability Index (VI) and light grey the lowest value. Data were reported as means of two independent experiments. 0.5 h and 2 h GJ = strain VI in Gastric Juice (GJ) after 0.5 h and 2 h of incubation, respectively; 3 h IF and 24 h IF = strain VI in intestinal fluid (IF)) after 3 h and 24 h of incubation, respectively.

**Figure 5 foods-11-01342-f005:**
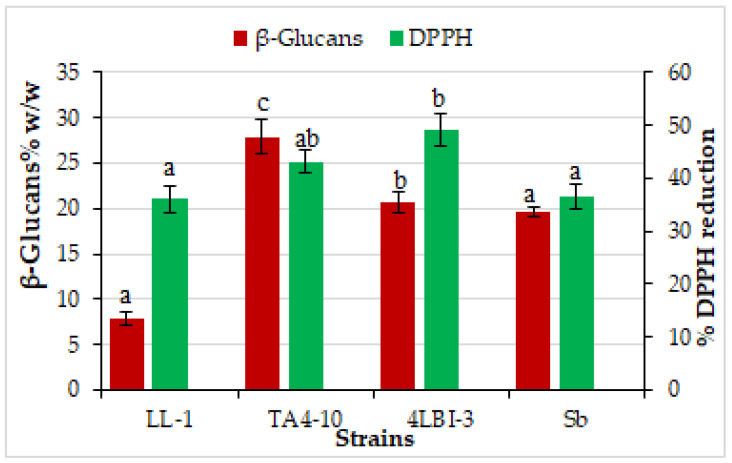
Determination of β-glucan content (%*w/w*) and antioxidant activity (%DPPH reduction) in four *Saccharomyces cerevisiae* strains. Letters on plot bars indicate significant differences (*p* < 0.05) among the four selected strains for each parameter. Data are expressed as mean value ± SD of two independent experiments.

**Figure 6 foods-11-01342-f006:**
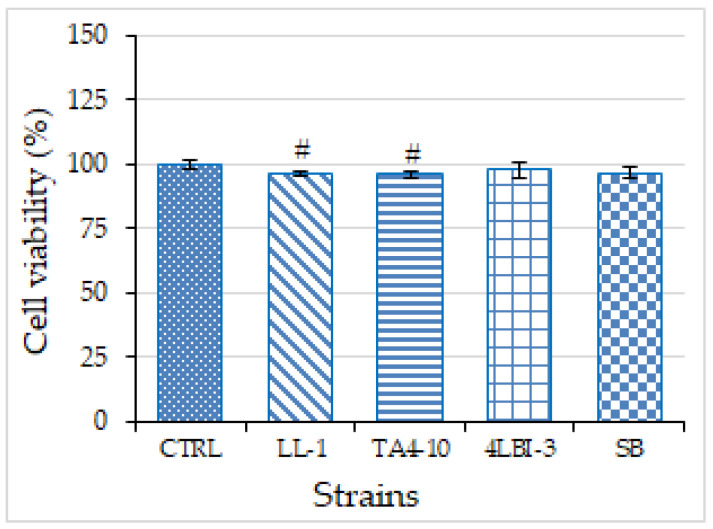
Effect of TA4-10, LL-1, Sb and 4LBI-3 on U937/PMA macrophage cell line viability. U937 cells were seeded in the presence or not (CTRL) of the four selected strains. Following 72 h cell viability was measured by cell counting. # *p* < 0.05 vs. CTRL (one-way analysis of variance, ANOVA).

**Figure 7 foods-11-01342-f007:**
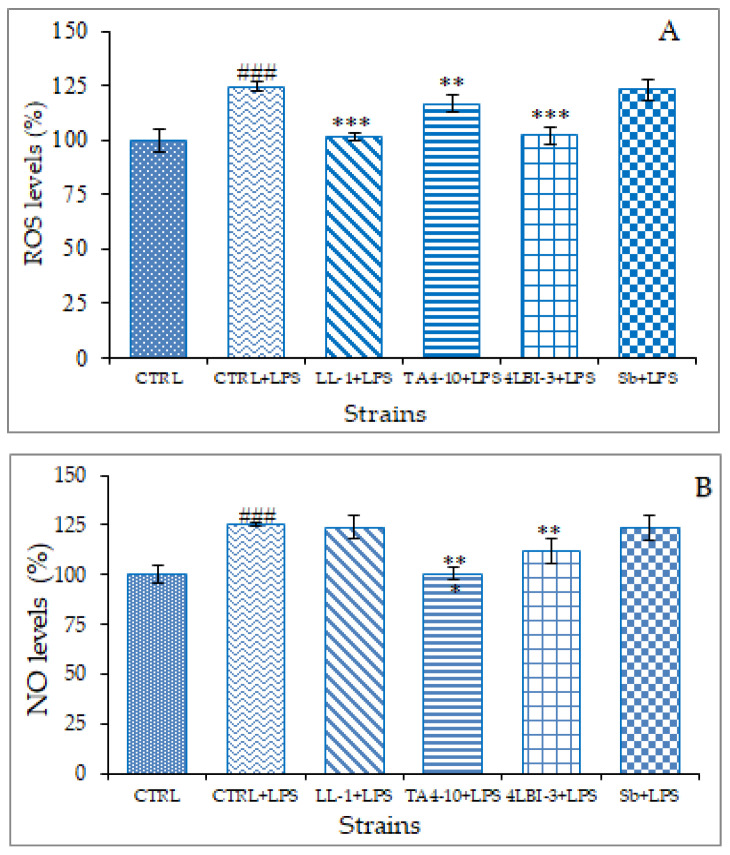
Anti-inflammatory activity of TA4-10, LL-1, Sb and 4LBI-3 strains in U937/PMA macrophage cell line. Anti-inflammatory effect of all four trains has been evaluated by measuring ROS (**A**) and NO production (**B**) in LPS-induced macrophage cell line. U937/PMA cells were triggered by 200 ng/mL of LPS in the presence or not (CTRL) of TA4-10, LL-1, Sb and 4LBI-3. Following 24 h ROS and NO levels were quantified. The symbol # means comparison vs. CTRL, whereas the symbol * means comparison vs. CTRL + LPS. ** *p*< 0.01; ### and *** *p* < 0.001, (one-way analysis of variance, ANOVA).

**Table 1 foods-11-01342-t001:** Origin of the fifty indigenous *Saccharomyces*
*cerevisiae* strains tested in this study.

Strain Code	Origin	Reference
AC3-1; TA4-10	Inzolia Grape Must, Sicily (Italy)	[41]
CA10-4sc2; CB1-3sv2; CB2-8sr3; CD2-6sc2; CD7-3sv2; TA8-4sc2; RA2-1sb2; RA8-1sv2; RE8-7sr3	Nero d’Avola Grape Must, Sicily (Italy)	[42]
ANF1-9; ANF8-3; ANFA-12; ANFB-11; ANFC-3; ANFC-30; ANFD-18; ANFD-21; GEO2-8	Kaketian Wine, (Georgia)	[43]
SA7-13; SA10-19; SB5-18; SC5-25; SC5-37	Sangiovese Grape Must, Tuscany (Italy)	[44]
BA-215; SN-41	Sangiovese Grape Must, Emilia Romagna (Italy)	[45,46]
BP2-33; Mpr2-18	Primitivo Grape Must, Basilicata (Italy)	[47]
E1-6; FIMA-3; M1-47; M1-110; M3-59; M3-60; M3-80; VEME-2; 4LBI-3	Aglianico Grape Must, Basilicata (Italy)	[45,48]
FD-3; LL-1; M5-15; P4	Sourdough, Basilicata (Italy)	[49]
M4	Beer, Basilicata (Italy)	[49]
5TB8-20; 5TB8-28; 5TB8-60; 5TB8-73	Bosco Grape Must, Liguria (Italy)	[50]
9-15b; 9-15G; 12-15b	Honey, (Malta)	[49]

**Table 2 foods-11-01342-t002:** Traits of the 14 selected strains. Data of growth at low pH (pH 2.5 and 3.0) after 24 h are expressed as Growth Index (GI, %). Hydrophobicity is expressed as MATS (%) and reported as mean value ± SD of two independent experiments.

Strains	GI (%) pH2.5	GI (%) pH3.0	MATS (%)
4LBI-3	50–75	>75	91.98 ± 10.25
BA-215	50–75	50–75	55.95 ± 5.80
CA10-4sc2	>75	>75	51.18 ± 4.99
CD2-6sc2	50–75	>75	52.51 ± 5.60
E1-6	50–75	>75	74.53 ± 7.50
EC1118	50–75	>75	10.19 ± 1.05
LL-1	50–75	>75	98.79 ± 9.60
M3-59	<50	>75	58.17 ± 4.84
M4	50–75	>75	57.98 ± 5.75
M5-15	>75	>75	78.50 ± 6.88
P4	50–75	>75	92.37 ± 8.98
Sb	<50	>75	36.48 ± 3.55
SC5-25	50–75	>75	59.30 ± 5.13
TA4-10	>75	>75	96.77 ± 9.42

## Data Availability

The data presented in this study are available in the article.

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
