# Peer review of "In Vitro Study of Probiotic, Antioxidant and Anti-Inflammatory Activities among Indigenous Saccharomyces cerevisiae Strains"

_foods, 2022, doi:10.3390/foods11091342_

Round 1

Reviewer 1 Report

In this manuscript, the authors try to screen for probiotic yeasts for the application in food industry. They tested the yeasts’ probiotic, antioxidant and potential anti-inflammatory activities in vitro. The authors finished a lot of work. 

The first paragraph is too long, the authors need to separate into several sections.

The use of some yeast strain name is not right. For examples, Line 64, Saccharomyces cerevisiae should use as S. cerevisiae. Line 66-67, Saccharomyces should be italic. Line 82, Saccharomyces cerevisiae should use as S. cerevisiae. Other similar mistakes are available in the manuscript. The authors need to check carefully.

Table 1, the three-line table is not standard.

In the materials and methos part, the methods sources are not cited. For example, 2.4.2, 2.4.3.1, 2.4.3.2, 2.4.3.3. The reason to select the materials and methods are not described.

No error bar in Figure 1.

Figure 2-3, Figure 5-7 are in low-quality. These figures directly cut from excel and they are in low-quality.

Author Response

The authors are thankful to the reviewer for the valuable suggestions, very useful for improving our paper.

Reviewer 1

Comments and Suggestions for Authors

In this manuscript, the authors try to screen for probiotic yeasts for the application in food industry. They tested the yeasts’ probiotic, antioxidant and potential anti-inflammatory activities in vitro. The authors finished a lot of work. 

The first paragraph is too long, the authors need to separate into several sections.

We modified the first paragraph, accordingly to reviewer’s suggestion.

The use of some yeast strain name is not right. For examples, Line 64, Saccharomyces cerevisiae should use as S. cerevisiae. Line 66-67, Saccharomyces should be italic. Line 82, Saccharomyces cerevisiae should use as S. cerevisiae. Other similar mistakes are available in the manuscript. The authors need to check carefully.

We checked carefully all the manuscript, editing the yeast strain name.

Table 1, the three-line table is not standard.

We modified the format of Table 1.

In the materials and methos part, the methods sources are not cited. For example, 2.4.2, 2.4.3.1, 2.4.3.2, 2.4.3.3. The reason to select the materials and methods are not described.

We added more information in the Material and methods sections, both as new references or improving the description of the used methods.

No error bar in Figure 1.

We added the error bar in the new version of Figure 1.

Figure 2-3, Figure 5-7 are in low-quality. These figures directly cut from excel and they are in low-quality.

We improved the image quality of all the figures, by hoping that now they meet a satisfactory quality.

Reviewer 2 Report

The manuscript entitled “In vitro study of probiotic, antioxidant and anti-inflammatory activities among indigenous Saccharomyces cerevisiae strains” screened out yeasts with probiotic properties from food matrix and analyzed a variety of probiotic properties, but there are still some problems to be solved in this manuscript.

Comment 1: Line 51 Reference 7 is not available, please provide a link to the full text. Line52-53 please quote relevant references. Line 66-69 please quote references supporting the author's view here, because some probiotics play a probiotic role by regulating the changes of host intestinal microorganisms through metabolites such as bacteriocin. Therefore, it is necessary to quote relevant reference to support what the author said. Some references for author concern are listed below:

Xu C, Ma J, Wang W, et al. Preparation of pectin-based nanofibers encapsulating Lactobacillus rhamnosus 1.0320 by electrospinning. Food Hydrocolloids, 2022, 124, 107216.

Cordonnier C, Thévenot J, Etienne-Mesmin L, et al. Dynamic In Vitro Models of the Human Gastrointestinal Tract as Relevant Tools to Assess the Survival of Probiotic Strains and Their Interactions with Gut Microbiota. Microorganisms 2015, 3(4), 725-745.

Mu Z, Yang Y, Xia Y, et al. Probiotic yeast BR14 ameliorates DSS-induced colitis by restoring the gut barrier and adjusting the intestinal microbiota. Food Function, 2021, 12, 8386-8398.

Comment 2: Line133-146, why not count the yeast directly, but using this method in the manuscript, the direct counting method may be more intuitive.

Comment 3: Line 161 Kirby Bauer method is used here. Why not detect MIC or use inhibition zone diameter to reflect the antibiotic resistance of the strain? This experimental method is used in some other studies.

Comment 4: Line180-183 According to the author, the method of simulating gastric juice and intestinal juice has been modified according to reference 50. The pH of simulated intestinal juice in reference 50 is 7.5. Why did the author adjust the pH to 8.5 in line 182. It is understood that the pH of the small intestine is usually 6-7.5. Is there any special reason why the pH of intestinal fluid is 8.5? This is confusing. Please explain.

Comment 5: Line184-196 In the process of simulating gastrointestinal tract, the author picked up 2 ml sample every 30 min, and then whether the corresponding solution was supplemented to keep the total volume of the solution unchanged and adjust the pH constant. Or the author just continues the simulated gastrointestinal experiment after picking 2 ml sample each time.

Comment 6: In 2.4.1, only DPPH radical scavenging activity is selected for the detection of antioxidant activity, which is incomplete. Increasing antioxidant indexes such as scavenging of hydroxyl radicals, scavenging capacity for superoxide anion free radical can make this part more complete.

Comment 7: Line285 Corresponding to question 2, if yeast is counted, it may be more intuitive to use survival rate.

Comment 8: Line338-346 As mentioned in question 3, why not display the data here? MIC or inhibition zone diameter can be presented here. Finally, mark the strain R/I/S to intuitively reflect the antibiotic resistance of the strain. Without data display, researchers cannot verify its authenticity.

Comment 9: Line 470 The comparison between the results of this manuscript and references 65 and 66 provided by the author is meaningless. Firstly, reference 65 does not involve the scavenging effect of DPPH in vitro, so how to compare it? Secondly, the method used in reference 66 is different from that in this manuscript. How can it be explained that the antioxidant activity of yeast in this manuscript is higher than that of other lactic acid bacteria? Is it reasonable to compare it through the data that cannot be statistically significant?

Comment 10: There are some errors in the format of references 6, 18, 47 and 61.

Author Response

The authors are thankful to the reviewer for the valuable suggestions, very useful for improving our paper.

Reviewer 2

Comments and Suggestions for Authors

The manuscript entitled “In vitro study of probiotic, antioxidant and anti-inflammatory activities among indigenous Saccharomyces cerevisiae strains” screened out yeasts with probiotic properties from food matrix and analyzed a variety of probiotic properties, but there are still some problems to be solved in this manuscript.

Comment 1: Line 51 Reference 7 is not available, please provide a link to the full text.

This reference is a book chapter, the link for full text is not available

Line52-53 please quote relevant references. Line 66-69 please quote references supporting the author's view here, because some probiotics play a probiotic role by regulating the changes of host intestinal microorganisms through metabolites such as bacteriocin. Therefore, it is necessary to quote relevant reference to support what the author said. Some references for author concern are listed below:

Xu C, Ma J, Wang W, et al. Preparation of pectin-based nanofibers encapsulating Lactobacillus rhamnosus 1.0320 by electrospinning. Food Hydrocolloids, 2022, 124, 107216.

Cordonnier C, Thévenot J, Etienne-Mesmin L, et al. Dynamic In Vitro Models of the Human Gastrointestinal Tract as Relevant Tools to Assess the Survival of Probiotic Strains and Their Interactions with Gut Microbiota. Microorganisms 2015, 3(4), 725-745.

Mu Z, Yang Y, Xia Y, et al. Probiotic yeast BR14 ameliorates DSS-induced colitis by restoring the gut barrier and adjusting the intestinal microbiota. Food Function, 2021, 12, 8386-8398.

We added the references suggested by the reviewer.

Comment 2: Line133-146, why not count the yeast directly, but using this method in the manuscript, the direct counting method may be more intuitive.

By considering that a high number of strains was analyzed in this step, we used the absorbance measurements as this method supply more standardized values than direct counting method.

Comment 3: Line 161 Kirby Bauer method is used here. Why not detect MIC or use inhibition zone diameter to reflect the antibiotic resistance of the strain? This experimental method is used in some other studies.

The diameter of inhibition zone was not indicated as all the tested strains were antibiotic resistant.

Comment 4: Line180-183 According to the author, the method of simulating gastric juice and intestinal juice has been modified according to reference 50. The pH of simulated intestinal juice in reference 50 is 7.5. Why did the author adjust the pH to 8.5 in line 182. It is understood that the pH of the small intestine is usually 6-7.5. Is there any special reason why the pH of intestinal fluid is 8.5? This is confusing. Please explain.

We agree with reviewer’s comment and we thank him/her for valuable comment, sorry it was a mistake. We corrected with 7.5

Comment 5: Line184-196 In the process of simulating gastrointestinal tract, the author picked up 2 ml sample every 30 min, and then whether the corresponding solution was supplemented to keep the total volume of the solution unchanged and adjust the pH constant. Or the author just continues the simulated gastrointestinal experiment after picking 2 ml sample each time.

Following the protocol reported by Picot and Lacroix (2004) (reference 54 in this manuscript), we just continue the simulated gastrointestinal experiment after picking 2 ml samples.

Comment 6: In 2.4.1, only DPPH radical scavenging activity is selected for the detection of antioxidant activity, which is incomplete. Increasing antioxidant indexes such as scavenging of hydroxyl radicals, scavenging capacity for superoxide anion free radical can make this part more complete.

In this manuscript the antioxidant activity of the strains was evaluated also by ROS and NO detection (see paragraph 2.4.3.3); this part was separated by DPPH radical scavenging activity as a different methodological approach was used.

Comment 7: Line285 Corresponding to question 2, if yeast is counted, it may be more intuitive to use survival rate.

Please, see the comment previously reported (reply to comment 2)

Comment 8: Line338-346 As mentioned in question 3, why not display the data here? MIC or inhibition zone diameter can be presented here. Finally, mark the strain R/I/S to intuitively reflect the antibiotic resistance of the strain. Without data display, researchers cannot verify its authenticity.

As previously reported, we did not display the data as all the strains gave the same results, which were antibiotic resistance.

Comment 9: Line 470 The comparison between the results of this manuscript and references 65 and 66 provided by the author is meaningless. Firstly, reference 65 does not involve the scavenging effect of DPPH in vitro, so how to compare it? Secondly, the method used in reference 66 is different from that in this manuscript. How can it be explained that the antioxidant activity of yeast in this manuscript is higher than that of other lactic acid bacteria? Is it reasonable to compare it through the data that cannot be statistically significant?

We modified the text, following reviewer’s suggestion.

Comment 10: There are some errors in the format of references 6, 18, 47 and 61.

We modified the format of the references.

Reviewer 3 Report

Dear  editor and author(s),

Manuscript "In vitro study of probiotic, antioxidant and anti-inflammatory  activities among indigenous Saccharomyces cerevisiae strains' the topic of manuscript is a very interesting area of research. However, the authors have failed to meet the high standards of publication in Foods. Overall, the work requires substantial changes. Some suggestions include:

1-Based on genetic tests, the yeast S. cerevisiae var. boulardii was converted to the yeast S. boulardii and became an independent species. Throughout the manuscript, the name of yeast must be amended.

2- The abstract of the manuscript needs to add some numbers of results, Please, Abstract should be rewritten.

3-The aim of study is unclear. Please write the aim of study ............etc.

4-Some methods of this manuscript need to add references.

5- The microbiology unit must be rewritten and corrected in all parts of the manuscript , (cells/mL correct to CFU/mL).

6-Page 4 line 155,  How many cells are in this volume?

7-Page 5 line 215, Determination of Glucans, this method is unclear.

8-Page 8 line 316, Figure 1 there is no designation for the x- and y-axis.

9-Figures 1, 2, 5 axes are not clear in the drawing, they must be redrawn.

10-Writing the references in the manuscript and citing to them does not match with the journal style.

11-The conclusions in the manuscript contain results; please rewrite it again.

12-Several notes are in the attached file

Author Response

The authors are thankful to the reviewer for the valuable suggestions, very useful for improving our paper.

Reviewer 3

Comments and Suggestions for Authors

Dear editor and author(s),

Manuscript "In vitro study of probiotic, antioxidant and anti-inflammatory  activities among indigenous Saccharomyces cerevisiae strains' the topic of manuscript is a very interesting area of research. However, the authors have failed to meet the high standards of publication in Foods. Overall, the work requires substantial changes. Some suggestions include:

1-Based on genetic tests, the yeast S. cerevisiae var. boulardii was converted to the yeast S. boulardii and became an independent species. Throughout the manuscript, the name of yeast must be amended.

We agree with the reviewer that this a debated and quite confused topic; in fact, in the literature discording papers are available. However, we decided to maintain S. cerevisiae var. boulardii for the following reasons:

As reported by Staniszewski, A.; Kordowska-Wiater, M. Probiotic and Potentially Probiotic Yeasts-Characteristics and Food Application. Foods 2021, 10, 1306:  “S. cerevisiae var. boulardii was originally described as a separate species - Saccharomyces boulardii - but rapid development of molecular phylogenetics in recent years has led to a change in its classification, as has happened with many yeast species, and it is currently classified as Saccharomyces cerevisiae var. boulardii”.

Moreover, in NCBI Taxonomy Browser (see the link reported below) we found that Saccharomyces boulardii is invalid name which “refers to a name not published in accordance with rules enumerated in the International Code for algae, fungi  and plants”

(Link:https://www.ncbi.nlm.nih.gov/Taxonomy/Browser/wwwtax.cgi?mode=Info&id=252598&lvl=3&lin=f&keep=1&srchmode=1&unlock#note1).

2- The abstract of the manuscript needs to add some numbers of results, Please, Abstract should be rewritten.

Following the reviewer’s suggestion, we modified the abstract, by adding some results.

3-The aim of study is unclear. Please write the aim of study ............etc.

We added the aim of the study in the final part of the introduction section.

4-Some methods of this manuscript need to add references.

We added more information in the Material and methods sections, both as new references or improving the description of the used methods.

5- The microbiology unit must be rewritten and corrected in all parts of the manuscript, (cells/mL correct to CFU/mL).

We used cells/mL as the microbial load was estimated by optical density measurements (which usually is correlated to cells/mL) and not as viable count on plate.

6-Page 4 line 155, How many cells are in this volume?

In this volume, there were about 1x107 cells/mL. We added this information in the text.

7-Page 5 line 215, Determination of Glucans, this method is unclear.

In the material and methods section, we added more details for determination of Glucans.

8-Page 8 line 316, Figure 1 there is no designation for the x- and y-axis.

We added this information in the figure 1.

9-Figures 1, 2, 5 axes are not clear in the drawing, they must be redrawn.

We have redrawn and improved the image quality of all the figures.

10-Writing the references in the manuscript and citing to them does not match with the journal style.

We checked carefully all the reference in the manuscript and modified those did not match with the journal style.

11-The conclusions in the manuscript contain results; please rewrite it again.

We rewrote the conclusions, by eliminating numerical results.

12-Several notes are in the attached file

We answered point by point in the pdf attached file.

Round 2

Reviewer 2 Report

In "3.3.3" section. Line 574: This result seems to come from Figure 5, but here the author writes Figure 4. Please check and correct this.

Author Response

The authors are thankful to the reviewers for the valuable suggestions, very useful for improving our paper.

In "3.3.3" section. Line 574: This result seems to come from Figure 5, but here the author writes Figure 4. Please check and correct this.

The correction was made, according to editors’ suggestion.

Reviewer 3 Report

Dear editors,

Authors did all necessary changes to improve the manuscript and now I recommend it for publication in the current form.

Author Response

The authors are thankful to the reviewers for the suggestion.